# Micro Orthogonal Fluxgate Sensor Fabricated with Amorphous CoZrNb Film

**DOI:** 10.3390/s25165022

**Published:** 2025-08-13

**Authors:** Kyung-Won Kim, Sung-Min Hong, Daesung Lee, Kwang-Ho Shin, Sang Ho Lim

**Affiliations:** 1Department of Material Science and Engineering, Korea University, Seoul 02841, Republic of Korea; kkw8952@keti.re.kr; 2Smart Sensor Research Center, Korea Electronics Technology Institute, Seongnam 13509, Republic of Korea; smhong@keti.re.kr (S.-M.H.); leeds@keti.re.kr (D.L.); 3Department of Information and Communication Engineering, Kyungsung University, Busan 48434, Republic of Korea

**Keywords:** orthogonal fluxgate, micro-fabrication, amorphous CoZrNb film, sensitivity

## Abstract

We successfully fabricated micro orthogonal fluxgate sensors using amorphous CoZrNb films. The sensor, measuring 1.5 mm × 0.5 mm, consists of three main parts: the conductor for excitation current flow, the magnetic layer sensitive to an external magnetic field, and the detection coil for measuring output voltage dependent on an external magnetic field. The magnetic layer forms a magnetically closed-circuit in the cross-section, which reduces reluctance and power consumption. Key fabrication challenges, such as poor step coverage and delamination, were effectively addressed by adjusting the sputtering angle, rotating the substrate during deposition, incorporating a Ta adhesion layer, and applying O_2_ plasma surface treatment. Optimal sensor performance was achieved by vacuum annealing the CoZrNb films at 300 °C under an applied magnetic field of 500 Oe. This process effectively enhanced magnetic softness and induced magnetic anisotropy, resulting in both very low coercivity (0.1 Oe) and a stable amorphous structure. The effects of operation frequency and the conductor width on the output characteristics of the fabricated sensors were quantitatively investigated. The sensor exhibited a maximum sensitivity of 0.98 mV/Oe (=9.8 V/T). Our results demonstrate that miniaturized orthogonal fluxgate sensors suitable for multi-chip packaging can be applied to measure the Earth’s magnetic field.

## 1. Introduction

Since the mid-2000s, there has been an increasing demand for magnetic sensors with high performance in terms of sensitivity and resolution, along with small size, for various applications including smartphones [1,2]. The parallel fluxgate (PFG) sensor, well known for its reliable magnetic detection performance, has long been a benchmark in magnetic sensing. However, its miniaturization is not easy due to its complex structure compared to other magnetic sensor types [3,4]. In contrast to modulation-type magnetic sensors such as Hall effect sensors and magnetoresistive (MR) sensors, PFG sensors necessitate both excitation and detection coils, leading to structural complexity and limitations in miniaturization. Miniaturization of magnetic sensors is especially crucial for smartphone applications, as smaller sensors generally lead to reduced costs and a more compact design. Moreover, for fluxgate sensors, power consumption is directly proportional to the volume of the magnetic core that needs to be magnetically excited, meaning that miniaturization can also result in reduced power consumption [5].

Several research groups have explored the use of Micro Electro Mechanical Systems (MEMS) technology to miniaturize PFG sensors, some of which were about 5 mm in length [6,7,8,9,10]. The International Technology Roadmap for Semiconductors (ITRS) emphasizes the importance of adding MEMS functionality to integrated circuits (ICs), positioning it as a key ‘More-Than-Moore’ technology [11,12]. As such, the demand for increasingly smaller MEMS-based sensors, including three-axis magnetometers, is expected to rise. This trend is driven not only by smartphones but also by the rapid growth of the internet of things (IoT) and wearable devices, where low power consumption and a small footprint are critical for market competitiveness.

In the sensor manufacturing process, the application of multi-chip packaging (MCP) [13,14] and complementary metal–oxide–semiconductor (CMOS) technologies to save space and minimize signal loss is becoming widespread. For a three-axis magnetic sensor packaged with an ASIC using MCP, the magnetic sensor element has to be of limited size to fit within the package. For instance, a magnetic sensor with temperature compensation and high-speed digital communication through a Serial Peripheral Interface (SPI) may need to be manufactured to a size of 2.2 mm × 2.2 mm, corresponding to a 3 × 3 grid of solder balls. In such cases, the magnetic sensor element must measure approximately 1 × 1 mm^2^.

Recently, the orthogonal fluxgate (OFG) sensor has garnered attention as a promising alternative, offering a simpler structure while maintaining performance comparable to PFG sensors [15,16,17,18,19,20]. Unlike conventional PFG sensors, which require an excitation coil as well as a detection coil, OFG sensors could be magnetically excited by passing AC current directly through a conductive magnetic core without an excitation coil. This single-coil design provides significant advantages in miniaturization compared to PFG sensors. Zhi S., et al. demonstrated a novel MEMS-type OFG sensor, fabricated using standard microfabrication techniques, which exhibited a maximum sensitivity of 575 V/T. The total area of the sensor was 9.4 × 6.4 mm^2^, including 1 × 1 mm^2^ electrodes [21]. Guo B., et al. also presented a micro-OFG sensor fabricated with an S-shaped excitation wire, surrounded by an electroplated Permalloy layer and a 3D micro-solenoid, which exhibited a sensitivity of 660.8 V/T in a ±150 μT linear range [22]. The size of the sensor was 5.8 × 2.8 mm^2^. Zorlu O., et al. fabricated a micro OFG sensor with a length of 1 mm using an electroplated copper excitation rod surrounded by a Permalloy layer, demonstrating a sensitivity of 0.51 V/T [23].

Despite these advancements, achieving the miniaturization of OFG sensors while maintaining sufficient magnetic detection performance remains a key challenge. For practical applications in ASIC-based micro-sensor signal conditioning, electronic noise levels are typically in the range of a few hundred nV [24]. Therefore, the magnetic sensor has to offer sensitivity in the range of 1 mV/Oe to resolve the Earth’s magnetic field with a resolution better than 14 bits. The sensitivity of an OFG sensor is influenced by the core volume, the number of turns of the detection coil, and the driving frequency [17]. Balancing miniaturization with adequate sensitivity remains a significant challenge in this research field.

The purpose of this study is to develop a micro magnetic sensor based on the OFG mechanism capable of detecting the Earth’s magnetic field with sufficient sensitivity. To ensure effective magnetization by the excitation current and external magnetic field, a CoZrNb amorphous film with optimized soft magnetic properties was used as the sensor magnetic core. The OFG sensor was fabricated using a well-defined surface micromachining process [25,26]. By overcoming a few fabrication issues, such as step coverage and delamination, we successfully manufactured micro orthogonal fluxgate sensors. We investigated its optimal excitation frequency and the core width dependence of sensitivity and linearity. This result underscores the potential of the micro OFG sensor for applications requiring precise detection of the Earth’s magnetic field.

## 2. Experimental Method

Amorphous CoZrNb films were deposited onto Si wafers that had been thermally oxidized to form a 300 nm thick SiO_2_ layer using DC magnetron sputtering with an alloy Co_79_Nb_14_Zr_7_ target. During deposition, the substrate was rotated at 12 rpm and cooled by circulating water. It is well known that substrate rotation during sputtering ensures uniform film thickness and composition [27]. Additionally, substrate cooling is essential to maintain a constant temperature, which preserves the amorphous state and ensures film reproducibility [28,29]. To enhance the magnetic softness of the films [30], heat treatment was conducted at 300 °C in a vacuum. During this process, a magnetic field of 500 Oe was applied to the films to induce magnetic anisotropy [31]. The magnetic properties of the films were evaluated through M-H loops measured using a Vibrating Sample Magnetometer (VSM). Magnetic anisotropy and coercive force (*H_c_*) were determined from the measured M-H loops.

Figure 1 shows a 3D schematic view of the micro orthogonal fluxgate sensor. The sensor is composed of a multilayer thin film structure. The detection coil adopts a flat solenoid configuration, with the upper and lower segments connected at both sides through vertical via holes to form a continuous winding. Further structural details are provided in Figure 2.

In the case of PFG, the magnetization of the magnetic film rotates due to the magnetic field generated from an excitation coil, and the dynamic change in magnetic flux generated by the rotation of the magnetization is detected using a detection coil. Since the magnetization of the magnetic film depends on the external magnetic field, the voltage measured in the detection coil also depends on the external magnetic field. On the other hand, in the case of OFG, there is no excitation coil. When the driving current is directly passed through the conductor, a magnetic field is generated in the circumferential direction of the conductor, and the magnetization of the magnetic film surrounding the conductor rotates due to this magnetic field. Since the magnetization of the magnetic film varies depending on the external magnetic field, when a detection coil is used to detect the change in magnetic flux, the output voltage also varies depending on the external magnetic field [15,16,17].

Figure 2 presents SEM images of the fabricated sensor, which measures 1.5 mm × 0.5 mm. The detection coil has a solenoidal shape with 37 turns. Both the line width and spacing of the detection coil are 10 μm. The length of the magnetic film is 1.2 mm, and the width is 110 μm.

The detection coil is positioned in the central region of the magnetic layer, marked as the effective region (ER; 760 μm) in Figure 2a. Since the magnetization of the magnetic film outside the ER does not affect the output voltage measured from the detection coil, only the ER influences the sensor output. This means that a sensor that exhibits the same performance can be manufactured with a length slightly longer than ER. However, in order to facilitate experiments such as handling the sensor, it was manufactured with a length of 1.5 mm. Figure 2b provides an enlarged view of the square region indicated by the black dashed line in Figure 2a. The widths of the magnetic layer and the insulating SiO_2_ layer are 110 μm and 90 μm, respectively. The conductor width in Figure 2 is 80 μm. To quantitatively assess the sensor’s performance in this study, we fabricated sensors with conductors of various widths (20–80 μm).

Figure 3 shows cross-sectional schematics of the OFG sensor. The cross-section locations, indicated by white dotted lines and labels a–c in Figure 2a, provide insight into the structural complexity of the sensor. Each cross-sectional schematic expresses a specific aspect of the multilayer structure. Figure 3a illustrates a side view of the micro OFG sensor. The conductor is embedded longitudinally within the magnetic film, while the magnetic layer remains open at both ends along the longitudinal direction. As shown in Figure 3b, the sensor’s cross-sectional structure consists of three main components: the conductor through which the excitation current flows, the magnetic layers whose magnetizations are influenced by the external magnetic field and excitation current, and the detection coil that detects an output voltage dependent on the external magnetic field. In Figure 3c, it can be seen that the magnetic film has a closed magnetic circuit structure surrounding the conductor. This structure has a small magnetic resistance in the circumferential direction, which helps to lower the exciting current flowing in the conductor and reduce power consumption.

The detection coil features a flat solenoid structure with a height of 2.6 μm and a width of 130 μm. It was fabricated using a Mo/Al/Mo multilayer film, with Mo/Al/Mo thicknesses of 30/200/30 nm, respectively. While the Al layer serves as the primary conductor, the Mo layers function as diffusion barriers and adhesion layers [32]. Additionally, the Mo layer in the Mo/Al/Mo structure prevents oxidation of the Al film during the PECVD process for forming the SiO_2_ insulating layer and protects the Al film from over-etching when SiO_2_ is patterned using reactive ion etching (RIE). To minimize electro-migration, the Al film in the Mo/Al/Mo structure was doped with 1% Si [33,34].

The magnetic layers and the detection coil are separated by SiO_2_, serving as an insulating layer. The magnetic layer forms a magnetically closed loop within the cross-sectional structure, as shown in Figure 3c. While OFG operation is possible with a single-layer CoZrNb film as the magnetic core carrying current, the cross-sectional structure used in this study offers a few advantages. Since the magnetically closed circuit exhibits a smaller magnetic reluctance, the magnetic film could be magnetized with a relatively smaller excitation current than a sandwich-shaped multilayer film. Additionally, this structure incorporates a Mo/Al/Mo conductor with higher conductivity than a CoZrNb single layer, resulting in lower resistance and reduced power consumption. In the cross-sectional design, the conductor, composed of a Mo/Al/Mo multilayer film, is centrally positioned.

Since our sensor operates at a frequency in the MHz band, we have to consider the eddy current that may occur in the conductive magnetic thin film. If eddy current occurs inside the magnetic film, the effective permeability of the sensor will decrease, which will not only lower the sensitivity but also increase the magnetic loss. In order to determine whether eddy current occurs, we could calculate the skin depth considering the thin film characteristics. Assuming that the relative permeability of the thin film is ~1000 and the resistivity of the thin film is 110 μΩ-cm, the skin depth is ~5 μm. The thickness of the magnetic film (500 nm) used to fabricate the sensor is sufficiently thin compared to the skin depth, so the generation of eddy currents inside the magnetic film could be ignored.

Figure 4 illustrates a cross-sectional view of the fabrication process of the micro orthogonal fluxgate. The substrate used was a 525 μm thick silicon wafer with a 300 nm thick SiO_2_ layer formed by thermal oxidation. The Mo/Al/Mo multilayer film was deposited using a DC magnetron sputtering and subsequently patterned via wet etching to form the lower part of the detection coil. A 1 μm thick photoresist pattern served as an etching mask during the wet etching process. On the top of the lower coil pattern, a 300 nm thick SiO_2_ film was deposited to form an insulator between the conductor and magnetic layer, as shown in Figure 4a. The SiO_2_ film was deposited using low-temperature PECVD with SiH_4_ and N_2_O as the reaction gases in a 2:9 ratio. The lower part of the magnetic layer, a 500 nm thick CoZrNb film, was deposited via DC magnetron sputtering and patterned using a lift-off process. After forming an insulator, 150 nm thick SiO_2_, over the lower magnetic layer, the conductor layer, Mo/Al/Mo, was deposited using DC magnetron sputtering and patterned accordingly. Another 150 nm thick SiO_2_ film was deposited and patterned over the conductor layer, as shown in Figure 4b. These insulator layers were essential to ensure that the excitation current flowed exclusively through the conductor, preventing unintended current paths through the magnetic film. The SiO_2_ surrounding the core center was selectively removed by RIE, as shown in Figure 4c. The upper part of the magnetic layer was subsequently deposited and patterned to form a magnetically closed circuit structure in the cross-section, as depicted in Figure 4d. A 300 nm thick SiO_2_ insulator was then deposited over the upper part of the magnetic layer, as illustrated in Figure 4e. Afterward, the upper part of the detection coil, composed of the Mo/Al/Mo multilayer, was deposited and patterned, as shown in Figure 4f. Finally, a 300 nm thick SiO_2_ passivation layer was deposited, as illustrated in Figure 4g, and patterned via wet etching to create contact holes for wire bonding, as shown in Figure 4h. Once the microfabrication process was complete, the sensor element was annealed at 300 °C in vacuum. During annealing, a 500 Oe magnetic field was applied in the transverse direction of the sensor.

The fabricated micro OFG sensor was then mounted onto a PCB and wire-bonded for measurement. A function generator was used to drive the sensor, and its output voltage was monitored using an oscilloscope. During operation, a 10 Ω resistor was connected in series with the sensor to measure the current by monitoring the voltage across the resistor. The sensor was driven with a sinusoidal excitation current (AC 50 mA with a DC bias of 25 mA) over a frequency range of 2 to 26 MHz to determine the optimal driving frequency. An external magnetic field was applied to the sensor using a Helmholtz coil, with the current supplied by a bipolar amplifier. The applied magnetic field ranged approximately ±26 Oe. The measured signals were acquired by a PC via GPIB communication with the oscilloscope.

## 3. Results and Discussion

### 3.1. Improvement of Magnetic Properties of Amorphous CoZrNb Films

The performance of magnetic sensors, including fluxgate sensors, is significantly influenced by the properties of the magnetic film. Magnetic softness (low coercivity, *H_c_*, and minimal hysteresis) and magnetic anisotropy directly affect sensor performance. The lower *H_c_*, the higher the resolution, and the magnetic field measurement range is determined by the magnetic uniaxial anisotropy of the magnetic film.

Figure 5 presents the measured M-H loops of a 380 nm thick CoZrNb film: (a) as-deposited and (b) annealed at 300 °C. Each graph displays two M-H loops measured parallel to the magnetically easy and hard directions, respectively. In Figure 5a, the observed anisotropy in the as-deposited film was not artificially induced but is believed to stem from environmental magnetic fields and/or substrate stress. The film was 10 mm × 10 mm in size. Given its dimensions, the in-plane demagnetizing field could be negligible. Although accurately calculating the demagnetization field is quite difficult, the demagnetizing factor of the film in-plane is expected to be below 5 × 10^−5^, referring to previous studies [35,36].

Figure 5b illustrates the M-H loops of the film annealed in vacuum at 300 °C. During the annealing process, a magnetic field of 500 Oe was applied. The easy axis was determined by the direction of the applied magnetic field during annealing. It is hard to say that 300 °C was the optimized annealing temperature. Our preliminary studies indicate that amorphous CoZrNb films exhibit optimal soft magnetism when annealed at approximately 400 °C. However, when CoZrNb films were micro-processed to fabricate sensor elements and integrated into multilayer structures, their magnetic properties deteriorated after annealing at 400 °C, negatively affecting sensor sensitivity and linearity. Therefore, the annealing temperature was limited to 300 °C to preserve the film’s soft magnetic properties after sensor fabrication.

It is well known that the soft magnetic property of an amorphous magnetic film could be improved through heat treatment [30,31]. The internal stress formed when an amorphous magnetic film is deposited onto a substrate causes poor soft magnetic properties, but this stress can be relieved through magnetic annealing in vacuum. To prevent oxidation of the magnetic film, heat treatment must be performed in vacuum or an inert gas (usually argon) atmosphere. In order to form uniaxial magnetic anisotropy in the magnetic film, a strong magnetic field, more than 10 times the saturation magnetic field, must be applied in the tangential direction of the film surface during annealing. The sensitivity and resolution of the magnetic sensor could be improved by the soft magnetic properties of the magnetic film, and the measurement bandwidth of the sensor is determined by accurate uniaxial magnetic anisotropy.

The saturation magnetization *M_s_* and uniaxial anisotropic field *H_k_* were determined to be 930 emu/cc and 11.5 Oe, respectively. The *H_c_* values were determined to be 0.1 Oe in the easy direction and 0.2 Oe in the hard direction. The film’s resistivity was 110 μΩ-cm, the same before and after annealing, indicating that the crystal structure remained amorphous even after annealing [37,38]. Typically, when an amorphous magnetic film crystallizes due to annealing, its resistivity decreases significantly. Maintaining an amorphous structure is essential, as amorphous magnetic materials inherently exhibit superior magnetic softness. For decades, amorphous magnetic materials have been recognized for their excellent magnetic properties due to their lack of crystal anisotropy [39,40,41]. Bulk amorphous magnetic materials are typically produced through rapid solidification, forming thin ribbons (tens of μm thick). Optimized amorphous ribbons can achieve extremely low *H_c_* values below 0.1 Oe [40]. However, as mentioned in the introduction, film materials with excellent soft magnetic properties are preferable for micro-magnetic sensor fabrication.

Jiang X.-D., et al. developed a thermal annealing method to nano-crystallize amorphous CoZrNb films [42], reporting excellent magnetic softness under optimal conditions, with a relative permeability of 5000 and an *H_c_* of 1.5 Oe. Takahashi T., et al. demonstrated that magnetic anisotropy (1 - 13 Oe) and *H_c_* (0.2–1 Oe) could be adjusted through magnetic field annealing [43]. Fujiwara Y., et al. also investigated the annealing effects on the magnetic softness of amorphous FeSiBNb films [44], finding that annealing at a specific temperature reduced *H*_c_ to below 0.2 Oe, indicating homogenization of internal stress distribution. Swan G., et al. investigated the variations in *H_c_* with film thickness in amorphous soft magnetic films (CoZrTb, CoZrNb, and CoZrMoNi), reporting minimum *H_c_* values for 200 nm thick films: 0.2 Oe for CoZrMoNi, 0.8 Oe for CoZrTb, and 1 Oe for CoZrNb [45]. Comparing these references with the results of this study, it could be concluded that the CoZrNb films fabricated in this study exhibit excellent magnetic softness.

### 3.2. Prevention of Disconnection and Delamination of the Magnetic Film

Figure 6a presents a cross-sectional image of the magnetic film layer, showing disconnection at the step edges. In a multilayer with a step cross-section structure, poor step coverage at layer interfaces can lead to disconnection of the film and mechanical damage at these edges. To prevent such problems, step coverage must be improved. Generally, step coverage can be enhanced by adjusting deposition parameters such as sputtering power, argon pressure, or substrate temperature to increase surface atom mobility. However, in this study, process conditions could not be altered to maintain the amorphous structure of the magnetic film. Instead, the sputtering system was modified to improve step coverage. The angle between the sputter gun and the substrate was adjusted to be 30°, and the substrate was rotated at 12 rpm during deposition. This approach effectively enhanced the film’s step coverage.

Beyond resolving the problem of film disconnection, the slope at step edges was also reduced. Figure 6b displays the cross-sectional image of the film after applying the improved fabrication process. This approach effectively mitigated disconnection at step edges.

Figure 7 shows cross-sectional images illustrating film delamination in the micro OFG sensor during heat treatment and improvements achieved through process modifications. Heat treatment typically releases internal stresses accumulated during film deposition. However, multilayer structures composed of metal and insulator layers with poor adhesion may delaminate due to differences in coefficient of thermal expansion (CTE), Young’s modulus, and Poisson’s ratio [46].

In Figure 7a, the black dashed line indicates the location delaminated after annealing. The magnetic layer and insulator on the upper right of the black dotted line were peeled off and disappeared, and a platinum layer (Pt) was applied for the focused ion beam (FIB) process performed for cross-sectional observation. To enhance adhesion, a 5 nm thick tantalum (Ta) film was introduced. Ta film forms a multilayer interface structure consisting of TaSi and TaO_x_ at the SiO_2_ interface [47], making it widely used as an adhesion and barrier layer in previous studies of Co-based amorphous thin films [48]. Additionally, an O_2_ plasma treatment was also applied to the substrate before magnetic film deposition. O_2_ plasma treatment generates hydroxyl (OH) radicals on the SiO_2_ surface, increasing hydrophilicity and surface activation energy [49], thereby improving adhesion between the magnetic film and the insulator layer. Figure 7b presents a cross-sectional image of the OFG sensor after implementing these process improvements. These modifications effectively eliminated film delamination during heat treatment. The following lists summarize the key points described above.

Summary of key issues:Disconnection of the magnetic film at step edges due to poor step coverage in the multilayer thin film structure.Delamination of the magnetic film from the SiO_2_ layer during heat treatment caused by poor adhesion between layers.Mechanical damage during etching due to a lack of process separation, resulting in damage to neighboring layers.

Improvement solutions:
Step coverage was improved by adjusting the sputtering gun angle to 30° and rotating the substrate at 12 rpm without altering the deposition conditions.Adhesion was enhanced by introducing a 5 nm Ta layer that forms stable Ta/TaO_x_ interfaces with the SiO_2_ substrate.O_2_ plasma treatment prior to film deposition increased surface hydrophilicity and activation, further improving adhesion.A 30 nm thick Mo layer served as both an adhesion and barrier layer, protecting the Al from oxidation and etching damage during the RIE process.Film delamination was prevented by controlling the heating and cooling rates at 3 °C/min to minimize thermal shock to the device.

### 3.3. Output Characteristics of the Fabricated OFG Sensor

Figure 8 illustrates the external magnetic field dependencies of the sensor’s output voltages. To determine the optimal operating frequency, frequency was also treated as a parameter. The micro-sensor used in this measurement had an 80 μm wide conductor. A key advantage of OFG sensors is their ability to operate in fundamental mode using a DC bias of suitable amplitude [50]. Typical fluxgate sensors, such as PFG sensors, detect the second harmonic component of the external magnetic field [51]. However, detecting the output in fundamental mode results in simpler circuit design and easier signal conditioning compared to second harmonic detection. When the magnetic film is saturated in the circumferential direction by a DC bias current, domain walls disappear, reducing noise caused by domain wall movement. In this study, the excitation current applied to the sensor consisted of an AC component of 50 mA and a DC bias of 25 mA. Considering the sensor’s width (80 μm), the circumferential magnetic field generated by the DC bias current was estimated to be ~1.5 Oe. Based on the results in Figure 5, the magnetization along the width direction (the easy axis) was expected to be saturated at 1.5 Oe, leading to the disappearance of domain walls. The external magnetic field (~10 Oe) at which the output voltage reached its maximum was nearly identical to the anisotropic field (~11.5 Oe) observed in the M-H loop, as shown in Figure 5. The largest output voltage is obtained when the external magnetic field and the anisotropic magnetic field are almost the same, because the static magnetic energy of the magnetic film is minimum, allowing the magnetization rotation to be easy. When the external magnetic field increases to more than 10 Oe, the magnetization rotation is suppressed by the strong external magnetic field, resulting in a decrease in the sensor output. The maximum sensor output voltage was 9.8 mV at 9.98 Oe when operated at 10 MHz.

As shown in Figure 8, the sensor output voltage peaked at 10 MHz and declined at higher frequencies. Typically, the output voltage of an OFG sensor increases proportionally with the excitation frequency [19]. However, the results in Figure 8 indicate that output voltage and sensitivity decreased beyond 10 MHz. This trend is more clearly illustrated in Figure 9.

Figure 9 shows the sensitivity of the sensors as a function of the operating frequency. The conductor widths of the sensors were 20, 40, 60, and 80 μm. The excitation current of the sensors was AC 50 mA and DC bias 25 mA. The sensitivity of each sensor increased proportionally with the operating frequency up to ~10 MHz. The output voltage measured in the detection coil of the sensor could be approximated using Faraday’s law [52], assuming the demagnetization field of the magnetic layer is negligible.(1)$$V\left(t\right)=NA\frac{dB\left(t\right)}{\mathrm{dt}}=NAH_{ex}\frac{d\mu \left(t\right)}{\mathrm{dt}}$$
where *N* is the turn number of the detection coil, *A* is the cross-sectional area of the magnetic layer, *H_ex_* is the external magnetic field (usually DC or low-frequency), and *B*(t) and *μ*(t) are the time-dependent magnetic flux density and permeability, respectively. According to Equation (1), the output voltage and sensitivity of the OFG sensor depend on the operating frequency, as well as the turn number of the detection coil and the cross-sectional area. However, there is an upper limit (~10 MHz) beyond which sensitivity no longer increases. This frequency limit is significantly lower than the *LC* resonant frequency (~58 MHz), which was calculated based on the coil’s inductance (~500 nH) and parasitic capacitance (~15 pF). This limitation is attributed to the inductance and capacitance of the connecting wire used for measurement, in addition to the detection coil itself. Kim Y.H., et al. reported the relation between the output property of the OFG sensor and the impedance of its detection coil when the sensor is connected to an instrument with a coaxial line whose length is too long to be neglected [19]. They pointed out that the impedance of the wire connecting the sensor to the measuring instrument had a significant effect on the output of a micro OFG sensor, and in particular, the frequency at which the maximum output is obtained may be more affected by the connecting wire than by the impedance of the sensor. Through this result, it can be expected that the optimal frequency will not change significantly even if the line width of the detection coil is reduced and the number of turns is increased. Therefore, it could be seen that it is possible to further increase the sensitivity through optimization by reducing the sensor connecting wire and increasing the turn number of the detection coil.

As shown in Figure 9, the wider the conductor width at the same frequency, the higher the sensitivity. It is considered that the sensitivity increases as the width increases because the amount of excited magnetization increases. This means the cross-sectional area, *A* in Equation (1), is larger in the sensor with the wider conductor. The sensitivities were 0.66, 0.8, 0.9, and 0.98 mV/Oe for conductor widths of 20, 40, 60, and 80 μm, respectively. However, the increase in sensitivity is not directly proportional to the conductor width. This is because the amplitude of current is the same in all cases, so a narrower conductor generates a higher current density. To quantitatively analyze whether the sensitivity is not directly proportional to the current intensity, we are performing a nonlinear FEM analysis that could describe the magnetization behavior of the magnetic layer of our micro OFG sensor. We plan to report these results in a future paper.

Figure 10 shows the dependence of output voltage on the external magnetic field at 10 MHz. The excitation current applied to the sensors was 50 mA AC with a 25 mA DC bias. The conductor widths were 20 μm (C20), 40 μm (C40), 60 μm (C60), and 80 μm (C80). The output voltage increased with conductor width: 7.5 mV for C20, 8.4 mV for C40, 9.2 mV for C60, and 9.8 mV for C80.

The magnetic field at which the maximum output voltage was observed decreased slightly with increasing conductor width: 11.36 Oe for C20, 10.44 Oe for C40, 10.2 Oe for C60, and 9.98 Oe for C80. This trend is qualitatively attributed to differences in current density. Although the conductor consists of a multilayer Mo/Al/Mo structure, most of the current flows through the Al layer, as Al is thicker (200 nm) than Mo (30 nm). Furthermore, the conductivity of Al ~3.5 × 10^7^ S/m is higher than that of Mo ~1.9 × 10^7^ S/m. Assuming that the current in Mo is negligible, the current density in the Al layer was calculated as follows: 10.67 × 10^9^ A/m^2^ for C20, 5.33 × 10^9^ A/m^2^ for C40, 3.55 × 10^9^ A/m^2^ for C60, and 2.66 × 10^9^ A/m^2^ for C80. As the current density increases, the excitation magnetic field in the circumferential direction increases, leading to an increase in circumferential Zeeman energy [53,54]. Consequently, the magnetization of the sensor with the narrower width becomes saturated at a higher external magnetic field in the longitudinal direction. Additionally, as the conductor width decreases, the nonlinearity of the output near 0 Oe becomes more pronounced. This trend is more clearly observed in Figure 11.

Figure 11 presents the output voltage derivative with respect to the external magnetic field, representing the sensor’s sensitivity as a function of the external field. This figure could also be used to evaluate the linearity of the sensor. As shown in Figure 11, the measurable magnetic field range varies depending on the conductor width, consistent with the trends observed in Figure 10.

Sensitivities near 0 Oe were observed to be 1.73, 1.83, 1.31, and 1.36 mV/Oe for conductor widths of 20, 40, 60, and 80 μm, respectively. Considering the average sensitivities (0.98, 0.9, 0.8, and 0.66 mV/Oe for conductor widths of 80, 60, 40, and 20 μm, respectively), the sensor with a conductor width of 80 μm exhibited the best linearity. Furthermore, as the conductor width decreased, linearity near 0 Oe tended to deteriorate. Due to the complexity of the sensor’s magnetic layer structure, identifying the exact cause of this nonlinearity is not possible at the moment. However, it is presumed that narrower conductors lead to more complex magnetization structures within the magnetic layer, resulting in nonlinear magnetization rotation in response to the external magnetic field.

Based on the results in Figure 10 and Figure 11, we could conclude that designing the conductor width as wide as possible is beneficial in terms of both sensitivity and linearity. Additionally, these findings suggest that implementing a feedback circuit would further improve the linear operation of the sensor. It is well known that feedback circuits in magnetic sensors enhance output linearity [55,56].

## 4. Conclusions

In this study, we successfully fabricated a micro orthogonal fluxgate (OFG) sensor using amorphous CoZrNb magnetic film. The sensor was 1.5 mm long but had an effective region (ER) of 0.76 mm, indicating that it could be manufactured in a size suitable for multi-chip packaging (MCP). Structurally, the sensor was designed with a multilayer magnetic core to reduce magnetic reluctance and power consumption. To obtain excellent magnetic softness and magnetic anisotropy, which significantly affect the sensor performance, while maintaining a stable amorphous structure, the CoZrNb thin film was subjected to an optimized vacuum heat treatment. We addressed critical fabrication challenges, significantly enhancing step coverage by modifying sputtering conditions, adjusting the sputtering angle, and employing substrate rotation, while simultaneously resolving delamination through a Ta adhesion layer and O_2_ plasma surface treatment. The fabricated sensor showed maximum sensitivity at an operational frequency of 10 MHz, achieving a maximum output voltage of 9.8 mV at 9.98 Oe. Wider conductor widths further enhanced sensitivity (up to 0.98 mV/Oe) and improved linearity, confirming that design parameters significantly affect sensor performance. We have successfully fabricated an OFG micro sensor that is small enough to fit into a multi-chip package with an ASIC. We believe that it will be fully applicable to smartphone applications if the sensitivity is improved by optimizing the turn number of the detection coil.

## Figures and Tables

**Figure 1 sensors-25-05022-f001:**
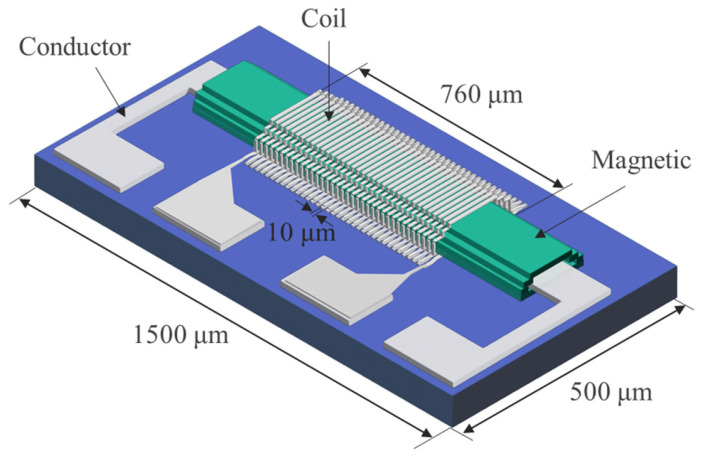
Three-dimensional schematic view of the micro orthogonal fluxgate sensor. For clarity, the insulating layers are not shown.

**Figure 2 sensors-25-05022-f002:**
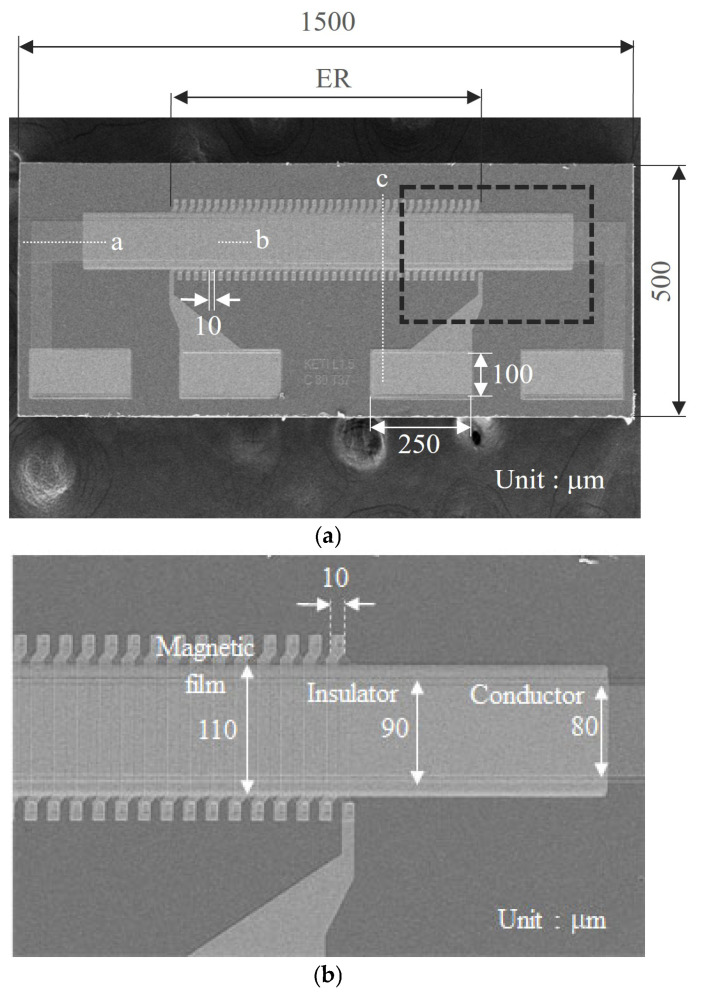
SEM images of the fabricated micro orthogonal fluxgate. The image shown in (**b**) is an enlargement of the square area surrounded by a dashed line box indicated in image (**a**). The dotted lines a–c shown inside (**a**) indicate the locations of the cross-sectional structures shown in Figure 3.

**Figure 3 sensors-25-05022-f003:**
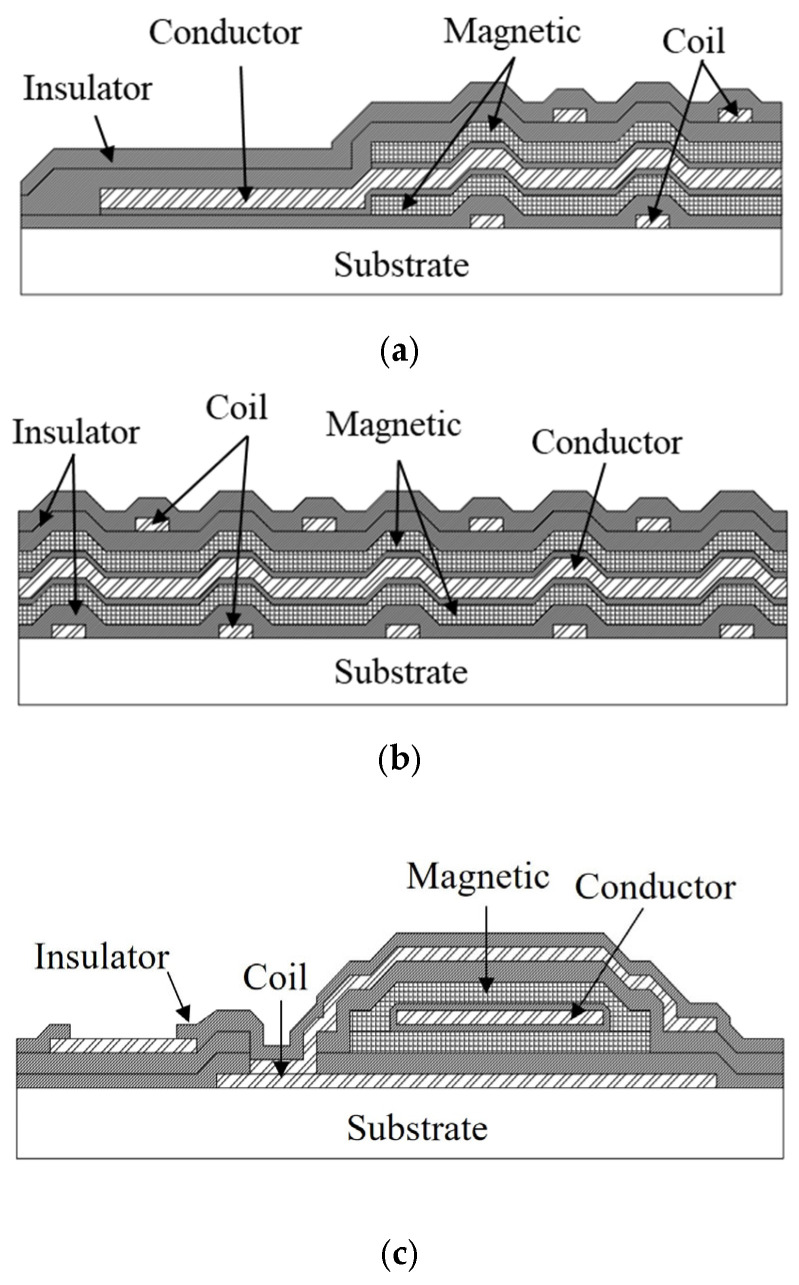
Cross-sectional schematic views. Cross-section cuts along the dotted lines indicated by (**a**–**c**) in Figure 2.

**Figure 4 sensors-25-05022-f004:**
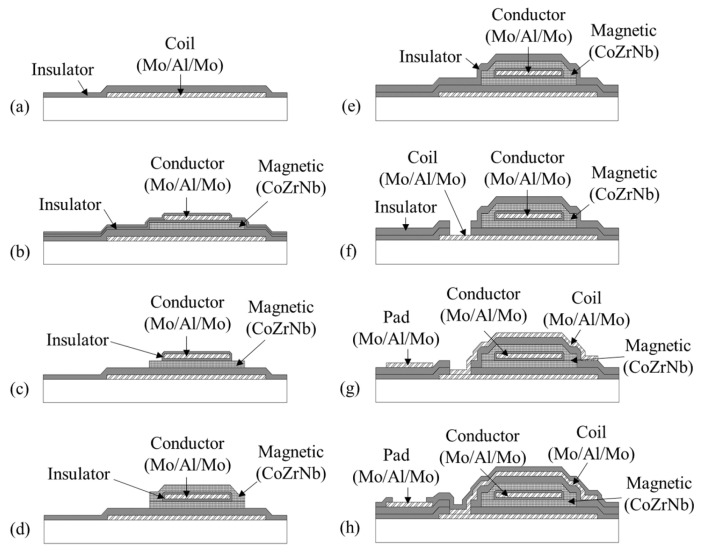
Cross-sectional schematic views of micro orthogonal fluxgate sensors for fabrication process. (**a**–**h**) shows each fabrication step once completed.

**Figure 5 sensors-25-05022-f005:**
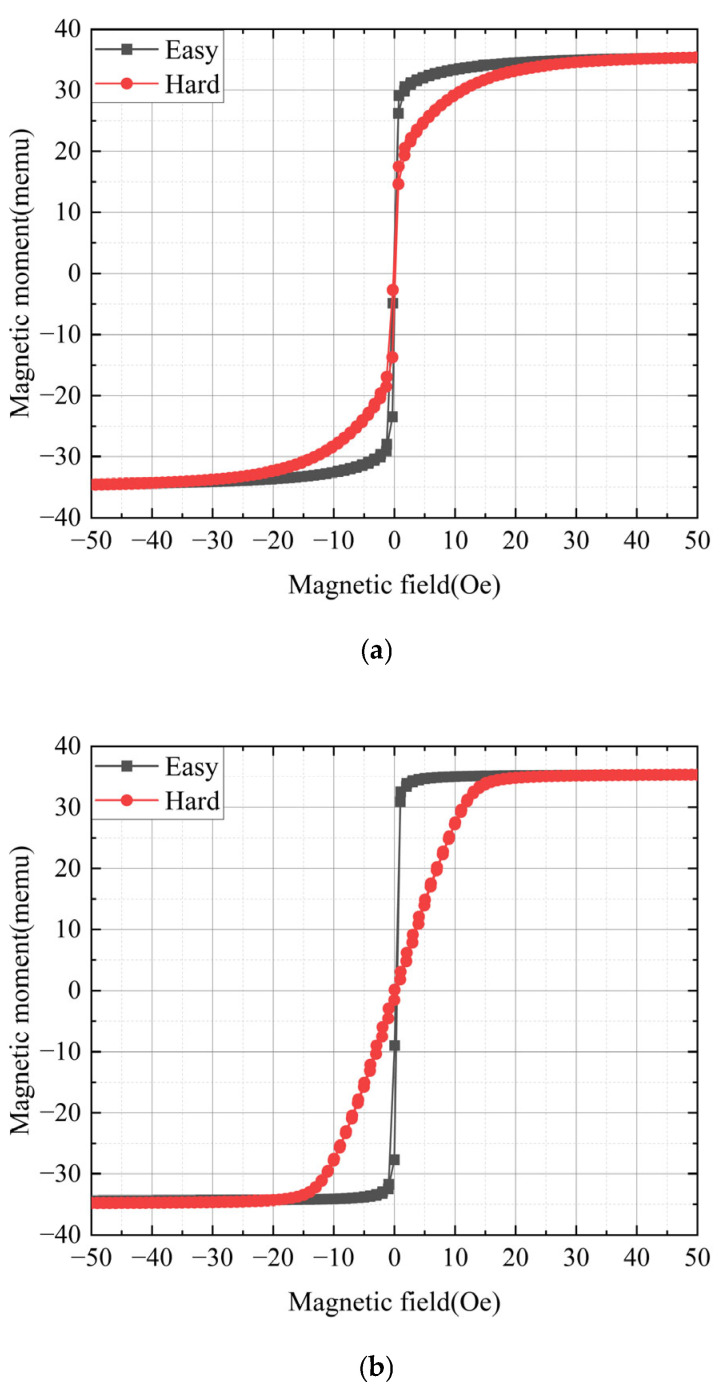
M-H loops of CoZrNb thin films: (**a**) as-deposited and (**b**) annealed at 300 °C.

**Figure 6 sensors-25-05022-f006:**
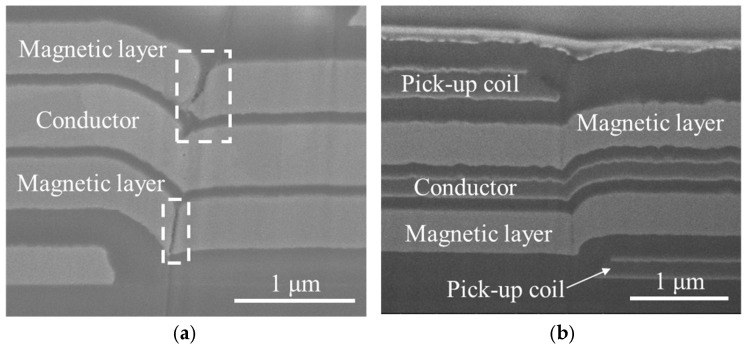
Cross-sectional SEM images of fabricated OFG sensors. The dashed white box indicates disconnected parts in the magnetic layer in (**a**); (**b**) shows the cross-section of the multilayer film fabricated after the disconnection issue was resolved.

**Figure 7 sensors-25-05022-f007:**
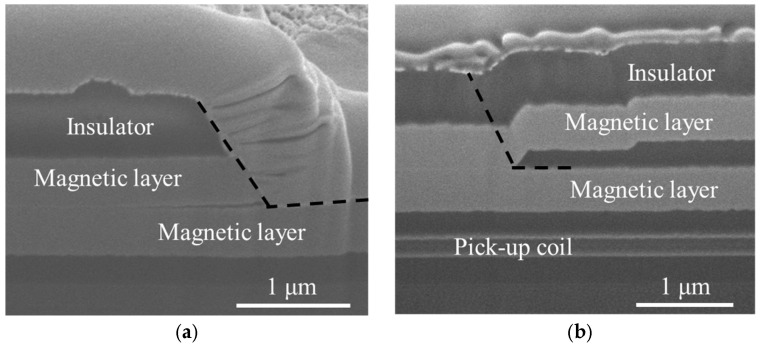
Cross-sectional image of the delaminated film (**a**) and cross-sectional image after improving adhesion of the magnetic layer (**b**). Delamination occurred due to poor adhesion between the magnetic film and the insulating layer during heat treatment. The black dashed line indicates the delaminated location.

**Figure 8 sensors-25-05022-f008:**
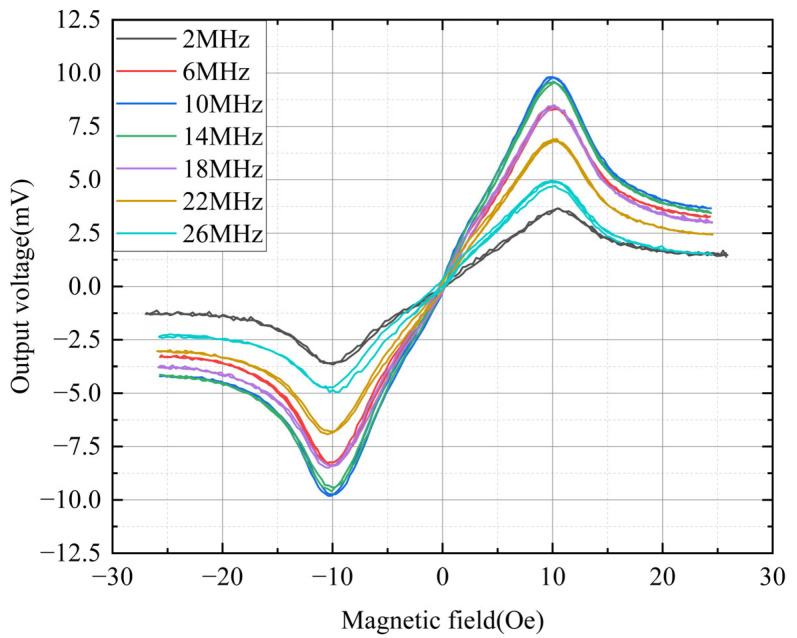
Dependence of output voltage on external magnetic field as a function of operation frequency from 2 MHz to 26 MHz.

**Figure 9 sensors-25-05022-f009:**
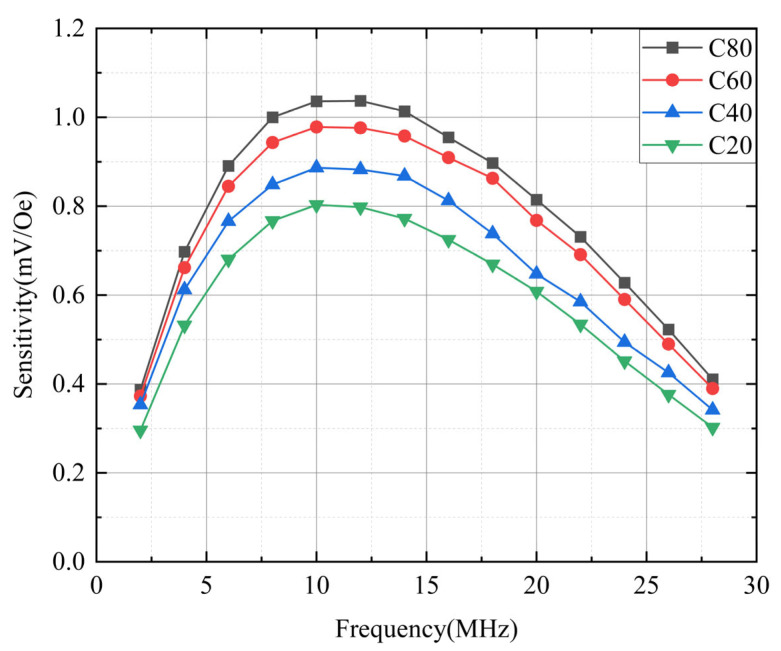
Frequency dependence of sensor sensitivity for conductor widths of 20 μm (C20), 40 μm (C40), 60 μm (C60), and 80 μm (C80).

**Figure 10 sensors-25-05022-f010:**
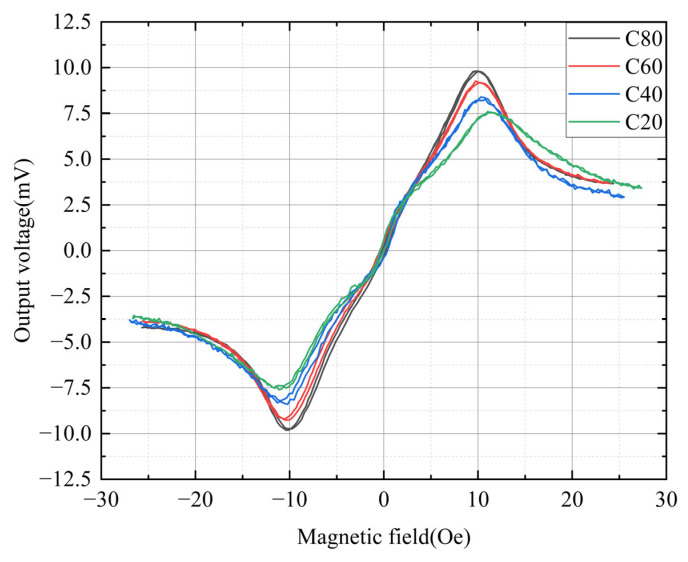
Magnetic field dependence of output voltages of the sensors for conductor widths of 20 μm (C20), 40 μm (C40), 60 μm (C60), and 80 μm (C80).

**Figure 11 sensors-25-05022-f011:**
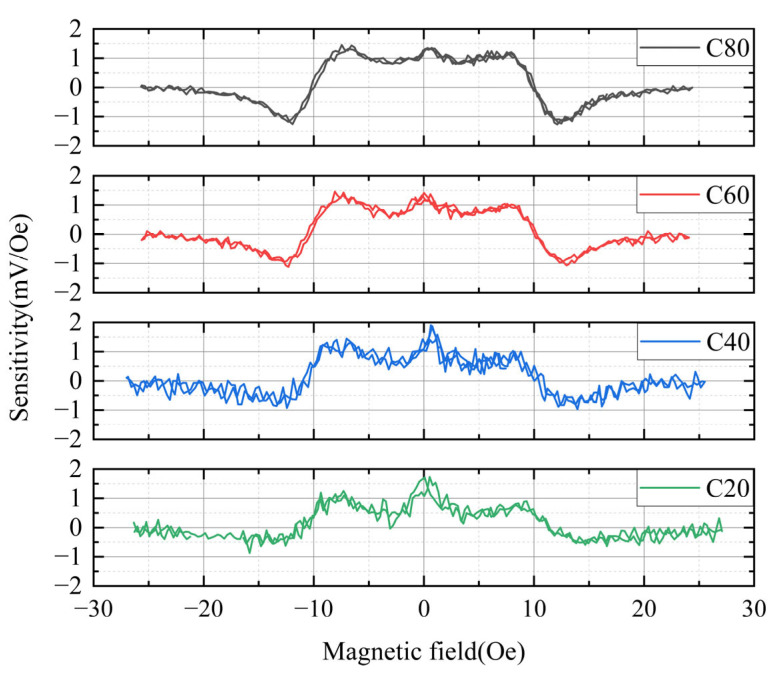
Dependence of sensitivity on external magnetic field. C20, C40, C60, and C80 represent conductor widths of 20 μm, 40 μm, 60 μm, and 80 μm, respectively.

## Data Availability

The data presented in this study are available on request from the corresponding authors due to the regulations of the research projects.

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
