# Peer review of "Micro Orthogonal Fluxgate Sensor Fabricated with Amorphous CoZrNb Film"

_sensors, 2025, doi:10.3390/s25165022_

Round 1

Reviewer 1 Report

Comments and Suggestions for Authors
  1. It could be better if add the sensor’s performance under the influence of different temperatures, thermal noise of the coil has a significant impact on the sensitivity of the sensor, and the coil structure of the device is relatively dense,
  2. How about the eddy current effect of magnetic cores?
  3. As i know, the micro-fluxgate including two coils, exciting coil and sensing coil, so how about the exciting method in this paper compared with the exciting coil method? Any more data to show?
  4. It could be better if add a picture to show the device structure in 3D view
  5. In sentence line 369-370, you point this “we are performing nonlinear FEM analysis that could describe the magnetization behavior of the magnetic layer of our micro OFG sensor. We plan to report these results in a future paper”, if possible, you can add some simulation in this paper to confirm this device design.
  6. It could be better if this paper can give a deeply comment about “vacuum annealing with magnetic field method” to the sensor performance

Reviewer 2 Report

Comments and Suggestions for Authors

In paper, the author provided OFG sensor using film. Structure and manufacturing are key in the section 2. Then, in section 3 important dimensions are presented, and the basis research using dimensional changes is adequately completed. I expect it to be a good paper if some improvements are made.

  1. If author add a 3D concept image of design and measurement setup to Figure 1, it will be easier for readers to understand.
  2. Although Fig2 provides many cross-section images, it lacks explanations while showing many feature structures. This is a description of Figure 2 from line 122. An enhanced explanation is needed for each of (a) to (c).

  3. In Fig3, which shows the manufacturing process, the images should include more detailed materials and methods. In addition, explanations are sometimes missing in the text. For example, I can't find explanation for Figure 3(b).

  4. Visualized figures and plots need improvement. For example, the text in the figures and plots is too small.

  5. Rename all sub-titles. In section 3.2, for example, change the title of the section to a valid solution element or important structure.

  6. Use bullet points in the 3.2 section to summarize how to resolve the issue. It seems author clear few issues by own solution. Listed parametric solutions are good to be readers.

  7. The author describes the results with graphs. However, the author should discuss the reasoning behind the graph in more detail.

  8. The paper needs a photo of the multi-chip with ASIC packaged item.

Comments on the Quality of English Language

I can't evaluate the English.

Round 2

Reviewer 1 Report

Comments and Suggestions for Authors all my question have been answered clearly, the paper can be accepted in Sensors

Author Response

We would like to thank the reviewer for his valuable comments and suggestions, which helped make our paper more understandable and clear to readers. We are confident that we could be able to improve the paper by modification according to his comments and suggestions.

Reviewer 2 Report

Comments and Suggestions for Authors

Author made good revision. However, the paper can be accepted after few modifications.

  1. Enhance abstract, introduction, and conclusion by adding few more sentences. So, I hope that this research will further appeal to industry and academia.
  2. But, delete sentences that repeat the same thing in the abstract, introduction, and conclusion.
  3. In the introduction, strengthen your bibliography by citing recently published papers. Broadening range of bibliography can provide deeper insights.
  4. Enlarge figures 8 to 11. Although this study focuses on the manufacturing process, the plots showing the measurement results should be enlarged to emphasize them.
